# Preparation and Characterization of Natural Bleaching Gels Used in Cosmetic Dentistry

**DOI:** 10.3390/ma12132106

**Published:** 2019-06-30

**Authors:** Amalia Mazilu (Moldovan), Codruta Sarosi, Marioara Moldovan, Filip Miuta, Doina Prodan, Aurora Antoniac, Cristina Prejmerean, Laura Silaghi Dumitrescu, Violeta Popescu, Anca Daniela Raiciu, Vicentiu Saceleanu

**Affiliations:** 1Physics and Chemistry Department, Technical University of Cluj-Napoca, 28 Memorandumului Str., Cluj-Napoca 400114, Romania; 2Department of Polymer Composites, Babes-Bolyai University, Institute of Chemistry Raluca Ripan, 30 Fantanele Str., Cluj-Napoca 400294, Romania; 3Physical Metallurgy, Department of Metallic Materials Science, University Politehnica of Bucharest, Bucharest 060042, Romania; 4Department of Pharmacognosy Phytochem Phytoterapy, Titu Maiorescu University, Faculty of Pharmacy, 16 Gh Sincai Str., 4 Dist, Bucharest 040405, Romania; 5Faculty of Medicine, University Lucian Blaga Sibiu, 2A Lucian Blaga Str., Sibiu 550169, Romania

**Keywords:** bleaching gels, SEM, biomaterials, dental color, digital image

## Abstract

The novelty of this study consists of the formulation and characterization of three experimental bleaching gels with hydroxylapatite oxides and fluorine (G28®, G29®, G30®) based on natural fruit extracts compared to the commercial Opalescence 15% (GC, Ultradent, South Jordan, UT, USA). Studies have been conducted on the effect that the experimental bleaching gels have on the color and morphology of different restorative materials (Nanofill®-Schulzer, P.L. Superior Dental Materials GmbH, Hamburg, Germany, and experimental nanocomposites (P11®, P31®, P61®)), immersed in coffee and artificial saliva (for 10 days and 30 days). The study also includes a cytotoxicity test on the gels and nanocomposites after bleaching, with ISO 109993-5 protocols on human dental follicle stem cells. UV-VIS spectroscopy, computerized measurement, and fluorescence spectrometry were used in order to observe the color changes, while the microstructure of the surface was investigated by Scanning Electron Microscopy (SEM). All of the samples immersed in coffee showed the highest color shift in comparison to the baseline. The color difference ΔE values obtained using the two methods (UV-Vis, computerized based on digital images) both after coloring and bleaching, respectively, were different for all four types of nanocomposites stored in the coffee, while no difference was observed in those stored in artificial saliva. The studied experimental gels and nanocomposites had a low cytotoxic effect on cell cultures after bleaching.

## 1. Introduction

Color anomalies are well-defined chapters in reference books written on dental aesthetics. This is due to fact that many monographs have been published on the subject of dental color [1,2]. Throughout the world, many studies have conducted analyses of dental changes caused by staining, and of possible therapies for color changes using non-invasive methods [3]. Current teeth whitening treatments usually use carbamide peroxide [4,5,6,7], hydrogen peroxide [7], or both [8,9]. Recently, agents containing enzymes for teeth bleaching have also been introduced [10]. Carbamide and hydrogen peroxide work the same way. Carbamide peroxide decomposes intraorally into hydrogen peroxide and urea, but the compound responsible for the bleaching effect is hydrogen peroxide. When the bleaching process is carried out, hydrogen peroxide operates by diffusion and enamel oxidation, breaking the bonds between colored staining molecules. Therefore, the stains remain on the teeth, but become invisible to the naked eye. The surfactants from the bleaching agent help detach and remove the stains from the extrinsic surface [11].

Considering the relationship between whitening materials and natural products, little is known about the possibility of obtaining a whitening effect using natural extracts.

The use of natural products in dentistry has been described in various situations, such as oral hygiene products (dentifrices, oral rinse solutions, restorative materials and materials for endodontics, periodontal dressing), cavity prevention [12,13,14], and xerostomia [15]. In 2000, Yu et al. [16] tested the in vitro capacity of the stain removal efficiency of two new fluoride dentifrices containing essential oils. In a 4-week clinical trial, in 2010 Xie et al. [17] tested the effect of toothpaste containing d-limonene on natural extrinsic smoking stains. For comparison and contrast purposes, the effects of d-limonene on tea stains were also assessed [17]. In another study, Kalyana et al. [18] tested the stain removal effectiveness by a novel dentifrice containing papain and Bromelain extracts. They concluded that there were significant results on stain removal with the new whitening dentifrice when comparing control samples. The determining role of the active drug substance in inducing dental dyschromia has been studied by Addy M. 1995 and Joiner A. 2003 [19,20]. Furthermore, the changes of dental color can constitute a diagnosis indicator for certain diseases [21,22]. There have also been studies about devising certain methods of chemical therapy for dental dyschromia with oxidative potential, associating them with testing of their local and systemic side effects [23,24,25].

Tao Jiang et al. evaluated in 2008 [26] the effect of a combination of hydroxyapatite (HA) and hydrogen peroxide (HP) on color, microhardness and morphology of human tooth enamel. The use of a 30% HP solution resulted in significant microhardness loss and morphological changes of enamel. HA could significantly reduce the microhardness loss of enamel caused by 30% HP and keep the enamel surface morphology almost unchanged. However, combination of HA and HP was not able to create a better whitening effect than HP alone. The HA could be a potential biomaterial used for teeth bleaching that ensures enamel protection. 

Jung-Hyun Son concluded that a whitening treatment applied on bovine teeth treated with whitening gel containing 35% hydrogen peroxide led to dramatically decreased enamel crystallinity, while professional whitening treatment with hydrogen peroxide combined with diode laser irradiation improved not only the whitening effect, but also protected the change of enamel structure in comparison to the whitening treatment without laser irradiation [27]. 

The novelty of this study is the formulation and characterization of new experimental bleaching gels, free of peroxides or any other chemical whitening agents, based solely on natural plant extracts. The aim of this research was to obtain superior properties such as: stability over time, ability to repair damage to the enamel structure after the bleaching process, the reduction of existing pigmentation and fast bleaching providing an optical brightening effect. 

The purpose of this paper was to study the effects of various bleaching agents with hydroxylapatite oxides and fluorine (G28, G29, G30) in comparison to a commercial bleaching gel (Opalescence 15% -GC) on the surface morphology and the color of different composite materials (Nanofill®, Schulzer and 3 experimental nanocomposites (P11, P31, P61)) and the study of the cytotoxicity of gels and nanocomposites after bleaching.

## 2. Materials and Methods

### 2.1. Experimental Bleaching Gels

For this study, three experimental bleaching gels, G28, G29, G30, were formulated and, for comparison purposes, a commercial gel (GC-Opalescence 15%) was used. The composition of the experimental whitening gels was: G-28: gout and strawberry juice, colloidal silica (Degusa), hydroxylapatite (HA*) and hydroxylapatite with oxides and fluorine (HA-ZnO_2_*, HA-F*); G29: quince juice, colloidal silica, hydroxylapatite (HA) and hydroxylapatite with oxides and fluorine (HA-ZrO_2_*, HA-TiO_2_*), G30: pineapple juice, colloidal silica, hydroxylapatite (HA) and hydroxylapatite with oxides and fluorine (HA-TiO_2_ HA-F*); Thickener agent: gelatine; Antimicrobial agent: salicylic acid (*synthesized in UBB-ICCRR-laboratory). Natural fruit juices were obtained by squeezing and concentrated by lyophilization (Lyophilizer—Model Alpha 1-4LDPLUS, Martin Christ Gefriertrocknungsanlagen GmbH, Osterode am Harz, Germany).

### 2.2. HPLC Analysis of Organic Acid Content

The analyses were carried out on a Jasco Chromatograph (Jasco International Co., LTD., Tokyo, Japan) equipped with an intelligent HPLC pump (PU-980, Jasco International Co., LTD., Tokyo, Japan), a ternary gradient unit (LG-980-02, Jasco International Co., LTD., Tokyo, Japan), an intelligent column thermostat (CO-2060 Plus, Jasco International Co., LTD., Tokyo, Japan), an intelligent UV/VIS detector (UV-975, Jasco International Co., LTD., Tokyo, Japan) and an injection valve equipped with a 20 μL sample loop (Rheodyne, Thermo Fischer Scientific, Waltham, MA, USA). Separation was carried out on a Carbosep Coregel 87H3 column (300 × 7.8 mm), Carbosep 87H guard column and Carbosep coregel 87H Cartridge at 35 °C column temperature. The mobile phase was the sulphuric acid 0.005M solution. The flow rate was 1 mL·min^−1^ and UV detection was 214 nm. The system was controlled, and the experimental data analysis was performed with the ChromPass software. 

*Sample preparation*: 1 g of bleaching gel was dissolved in water (up to 10 mL in a volumetric flask); the solution was sonicated for 15 min and then filtered under vacuum on a 0.45 μm filter and injected into HPLC. The acids required to perform the calibration curve: tartaric and citric acids were purchased from Merck (Darmstadt, Germany), succinic and oxalic acids were purchased from Reactivul Bucharest (Bucharest, Romania), and malic and fumaric acids were purchased from Polynt (Italy). The water used to prepare the standard solution and the gel extracts was Millipore water (18.2 M·cm^−1^). For the construction of the calibration curve, a standard mixture of organic acids was used at different dilutions (7 concentrations) ranging from 25 to 1000 μg/mL for oxalic, citric, tartaric, succinic acids; 49.72–994.30 μg/mL for malic acid and 0.28 and 5.70 μg/mL for fumaric acid. The HPLC method was validated in [28].

### 2.3. Experimental Composites for Restoration

Three experimental composite biomaterials (P11, P31, P61), with the composition shown in Table 1, and a commercial Nanofill® (Schulzer, P.L. Superior Dental Materials GmbH, Hamburg, Germany) composite were studied. The light-cured system of the composites used for this study contained 0.5% camphorquinone (CQ-Merck, Darmstadt, Germany) photosensitizer and 1% amine dimethylaminoethyl methacrylate (DMAEM-Sigma-Aldrich Chemie GmbH, Steinheim, Germany).

*Sample preparation*: We selected the above-presented composite materials (Table 1) in order to carry out the experiments becuase they can be used for both anterior and posterior tooth restoration. Each type of composite resin sample prepared in Teflon molds (2 × 1.5 mm) was further divided into two groups (n = 20 were placed in artificial saliva and the other groups of n = 20 were placed in coffee for 10 and 30 days and bleached with experimental and commercial gels.

*Whitening protocol*: With the help of bonding applicators, whitening materials were applied in a thin and even layer of about 0.5 mm on all sample surfaces of composite materials. The whitening gels were kept on the composite samples for 4 h a day for a total of 5 days. After each session, whitening gels were washed under running water for 1 min, then kept in distilled water.

### 2.4. Determining of the Color Changes of the Restoration Materials after Application of the Investigated Bleaching Gels

Samples prepared in accordance with Section 2.3 were immersed in dyes, and measured after 10 and 30 days, respectively, in order to determine the color difference ΔE with the UV-VIS spectrophotometer (UNICAM-UV4, UNICAM LTD., London, UK) equipped with a RSA–UC–40 (Labsphere, UNICAM LTD., London, UK) integrative sphere. The reflection spectrum was recorded in the 380–770 nm range compared to a spectralon standard. All recorded data was measured in CIELAB coordinates (L*, a*, b*), using a C-type illuminator with 8° sphere geometry. Color changes (ΔE) for both methods were determined using the following equation:
ΔE = [(ΔL*)^2^ + (Δa*)^2^ + (Δb*)^2^]^1/2^(1)

*Computerized measurement*: For comparative studies, the “Dentcolor” computerized measurement of digitally captured images was used on initially colored, then bleached samples. The “Dentcolor” application aims to measure and display the color values in a digital image as the main objective, then for two colors specified by the user the color difference ΔE *.

### 2.5. Fluorescence Spectroscopic Evaluation of Composites before and after the Bleaching Process

This study determined the fluorescent emission of the experimental resin composites before and after addition of a natural whitening agent G28. Evaluation by fluorescence spectrometry was done on samples prepared according to Section 2.3 using an ABL & Jasco V 6500 spectrofluorometer (Jasco International Co., LTD., Tokyo, Japan) with 150 W xenon lamp under the following conditions: tape with (Ex) = 1 nm, tape with (Em) = 10 nM, response: 0.5 s, sensitivity: medium, measuring range: 200–900 nm, data pitch >2 nm, wavelength excitation: 200 nm, scanning speed: 2000 nm/min.

### 2.6. Microstructural Analysis of Composites by SEM

The morphology of the surface of the experimental composite biomaterials before and after application of the three experimental whitening gels (G28®, G29®, G30®, UBB-ICCRR, Cluj-Napoca, Romania) and the commercial gel (Opalescence 15%—GC, Ultradent, South Jordan, UT, USA) was monitored by electronic scanning microscopy (SEM-Inspect S, FEI Company, Hillsboro, OR, USA).

### 2.7. Biological Assays

*Cell source*: Human dental follicle stem cells, prepared as previously described by Lucaciu [27], were a generous gift from Olga Soritau PhD, Institute of Oncology, Cluj-Napoca, Romania. Cells were cultured in Dulbecco’s modified Eagle’s medium (DMEM), supplemented with 5% fetal calf serum, 50 μg/mL gentamicin, and 5 ng/mL amphotericin (Biochrom Ag, Berlin, Germany). Cell cultures in the third-fourth passage were used.

*Conditioned medium*: The culture medium conditioned with the experimental samples was obtained complying with the ISO 10993–12:2012 standard.

*Bleaching gels*: 0.2 g of each bleaching gel per 1 mL of culture medium was incubated for 30 min at room temperature and immediately after, the conditioned medium was sterile filtered and used for cell culture exposure. Dilutions were made using cell culture medium [29,30]. Bleached composites: each composite sample was incubated with cell culture medium (3 cm^2^/mL) for 24 h, and 72 h at 37 °C, afterwards; the medium was immediately used, undiluted for cell treatment. 

*Viability assay*: Human dental follicle stem cells were used. The cells seeded at a density of 10^4^/well in ELISA 96-well micro titration flat bottom plaques (Techno Plastic Products AG, Trasadingen, Switzerland) were settled for 24 h. Then, cells were exposed to the conditioned medium of each gel, prepared as described above, in increasing dilutions (0.01–0.0001), as well as the undiluted composites for 24 h. Viability was measured by colorimetric measurement of formazan, a colored compound generated by viable cells using CellTiter 96® AQueous Non-Radioactive Cell Proliferation Assay (Promega Corporation, Madison, WI, USA). Untreated cultures exposed to medium were used as controls. 

*Statistical method*: The statistical difference between the treated and control groups was evaluated by paired Student’s *t*-test and one-way ANOVA, followed by Bonferroni posttest using GraphPad; the results were considered significant at *p* ≤ 0.05. The statistical package used for data analysis was Prism version 4.00 for Windows, GraphPad Software, San Diego, CA, USA.

## 3. Results

### 3.1. Determination of Organic Acid Content (mg/g) in Bleaching Gels by HPLC Analysis

The HPLC chromatogram of the standard organic acid mixture is shown in Figure 1a, and the chromatograms of the bleaching gels are shown in Figure 1b. The organic acid contents (mg/g gel) in the bleaching gels according to HPLC analysis is shown in Table 2.

The organic acid content in the studied bleaching gels ranged between 19.3294 mg/g for GEL 29 (based on quince natural juices) and 39.8629 mg/g for GEL 28 (based on pineapple and quince natural juices). After examining the results presented in Table 2, we can assume that the malic acid has a decisive role in the bleaching process, with its concentration varying between 14,146 mg/g for GEL 29 and 33.39 mg/g for the GEL 28 [30,31,32,33].

### 3.2. Determining the Color Change after Applying the Whitening Gels by Computerized Measurement

The color change was investigated on digitally captured images on the initially colored and then bleached samples. Figure 2 shows digital images used for the color difference (ΔE_ab_) by applying the original software (Dentcolor application, UBB, Cluj-Napoca, Romania) [34].

In Figure 3 and Figure 4, one can observe the graphical representation of the color changes on P11, P31, and P61 restorations (experimental), as compared to Nanofill (commercial), by coloring them in coffee (for 10 and 30 days, respectively) and than whitening with G28, G29, G30 experimental bleaching gels, compared to GC (Opalescence 15%) comercial gel, recorded using the Dentcolor program for digitized images and a UV-VIS spectrophotometer.

The ∆E_ab_ color difference values can provide a clear picture of the “match” between two samples or dental units. A “0” value of ∆E_ab_ means that the two values are actually the same. Increasing the ∆E_ab_ values mean an increase in difference. From a clinical point of view, depending on the visual perception of the individuals and the environment, some thresholds have been set, namely the 50/50% perception threshold ∆E_ab_ = 1.0, and the acceptability threshold ∆E_ab_ = 2.7 [35]. The 50/50% perception threshold ∆E_ab_ = 1.0 represents the limit above which 50% of the observers will see a color difference, while the other 50% will not. The acceptability threshold ∆E_ab_ = 2.7 is the limit value at which patients will consider the color difference to be clinically acceptable.

Both graphical representations include the mean values of ΔE after staining (after 10 and 30 days immersion in the dyeing liquid) and the mean values of ΔE at 30 days after discoloration (after bleaching with natural gels G28, G29, G30 and GC commercial gel).

### 3.3. Fluorescence Spectroscopic Evaluation of Composites before and after the Bleaching Process

Fluorescence is an optical property defined as a physical phenomenon occurring in less than 10^−8^ s, where the material absorbs the UV light invisible to the human eye and emits visible light, especially in the blue spectrum [36]. The fluorescence emission spectra for the four investigated composites before and after application of the G28® experimental bleaching gel are shown in Figure 5. One can see that for all studied samples, the fluorescent emmision increased after bleaching. 

### 3.4. Microstructural Analysis of Samples of Composite Materials before, after Staining, and after Bleaching

The surfaces of the samples of the experimental biocomposites (P11, P61) and commercial composite (Nanofill) before and after application of experimental bleaching gels (G28, G29, G30) and commercial gel (Opalescence 15%), was monitored by electronic scanning microscopy (SEM, Inspet S, FEI).

The surfaces of the samples used to determine the color change (Section 3.3) were examined. The composite samples P11, P31, P61, and Nanofill were analyzed after they have been kept in artificial saliva and coffee for 30 days. Changes to the sample surface after bleaching were observed with the four gels taken into the study being used, and the images are presented in Figure 6.

One can see from Figure 6 that in the case of sample P11 stored in artifficial saliva, the bleaching in gel G28 determined no obvious changes of the sample’s surface. The same conclusion can be extended in the case of sample P61 coloured in coffee and bleached with gel G29. In the cases of samples 61 stored in artificial saliva, bleached with G30 gel and NS stored in coffee and bleached with GC, the bleaching process seems to have determined the formation of smoother surfaces.

### 3.5. Biological Assays

The evaluation of the cytotoxicity in the experimental gels as compared to the commercial gel was carried out in triplicate. The data was statistically analyzed, and the results were considered significant at *p* ≤ 0.05. For the statistical analysis of viability, we compared OD_490_ from each experimental group, studied at a certain time, with the initial value. Figure 7 and Figure 8 present the cell viability for the four tested gels on composite materials after bleaching with G28 gel.

The cell viability test shows no cytotoxic effects in the case of the experimental bleaching agents. Gel 28 exhibited a dose-dependent increase in cell viability. In the case of GC (Opalescence 15%) gel, one can see a decrease of the the cell viability at concentrations of 0.01% and 0.001% (Figure 7) in a dose-dependent manner.

## 4. Discussion

Although the composition and structure of the composite materials used in dentistry are very different from the compostion and the structure of the tooth, they must recreate a physiognomic perception similar to that of teeth. The artistic ability of the phisician plays an important role in the rendering of an aesthetic apparance to restorations, but the quality of restorative materials, the understanding of their behavior, can greatly benefit or constrain this role. The change of color of the restoration materials is a controversial phenomenon, being assigned by some authors to the category of adverse effects, as the degree of discoloration is not equivalent to that of dental color changes; although the color of the composites changes under the action of the whitening agent, studies show that their translucency does not suffer. In addition, it is recommended that whitening tratments should be applied before any direct or indirect dental restorations. The whitening treatment can be applied after a few weeks of the chemical treatment with peroxides, so that the dental color is stabilized.

Regardless of the factor triggering the chromatic change (thermal, chemical, radiation effects) and the mechanism through which it proceeds (resin transformation or stains absorption), the quantitative evaluation methods are based on color recordings on samples before and after a simulated “in vitro” effect.

Material scientists have been inspired by the combination of hard (HA) and soft (collagen), and have carried out studies in order to formulate and develop new polymeric nanocomposites with the potential to be used as biomaterials [37]. 

There is research on the effects of cationic or anionic substitutions in the hydroxyapatite matrix on its biofunctionality in biomaterials [38]. Hydroxyapatite doping has been shown to result in many beneficial effects, such as: increased biological activity, cytocompatibility, cell viability and proliferation, adhesion, hemocompatibility, or antimicrobial or antifungal activity. However, it is necessary to compare the studies conducted in order to determine the best procedures to follow in order to determine exactly what causes the beneficial effects of substituted hydroxyapatite with various cations or anions [39]. 

The results show that the composition of the investigated experimental gels depend on the type of natural extract used to obtain them.

Organic acids, used as active agents, play an essential role in tooth discoloration and stain removal. To remove a possible negative effect of etching the enamel surface after repeated bleaching processes, hydroxyapatite particles substituted with various cations or anions have been added, having remineralization, anti-inflammatory and bacterial protection roles. 

According to Bartek [40], a manufacturer who develops chemical cosmetics and foods, malic acid is more balanced than other fruit acids, with a better buffering capacity than citric acid and lactic acid. Natural gels are currently a more desirable and less aggressive tooth whitening option, with this class of materials being under-studied so far.

Enzymes in the heminic peroxidase class use hydrogen peroxide to generate strong oxidizing agents, especially high-valency iron. Under weak acidic conditions but in the absence of hydrogen peroxide, peroxidases are also known to engage in a pseudo-oxidative reaction in which the first phase of reducing agents in the medium (feasible and saliva) reduces Fe (III) from peroxidase to Fe (II), then this Fe (II) reduces the molecular oxygen to the peroxide, after which the formed peroxide generates with the peroxidase. Thus, peroxidase can oxidize some organic substrates without the addition of hydrogen peroxide, as long as the medium is not completely anaerobic. The primary role of the biocatalyst is to generate a free radical at the unwanted colored compound, in the idea that after successive transformations of this type, the compound will be hydroxylated and passed into an oxidized, colorless quinone form.

Experimental composite P61 is more colored than Nanofill P11 and P31 composites immersed in coffee. After bleaching with both natural and commercial bleaching gels, the resin composites were whitened. The effects of bleaching on composite materials vary depending on the resin and composition of the bleaching gel, and the frequency and duration of exposure. Visually, the composite materials suffer a more intense coloring after 30 days, but after the whitening gel is applied, obvious discoloration of the samples occurs. It was found that the bleaching agents used behaved differently for the different composites studied. 

The results showed that the average color difference ΔE significantly decreased after whitening, for all bleaching gels in the two storage environments (coffee and artificial saliva). These color changes may be related to differences in chemical composition, particle size, and filler distribution.

The immersion time and the composition of the gels were significant factors that strongly influenced the bleaching process. The correlation between the immersion time and the type of bleaching gel was also significant. The ΔE values (Figure 3 and Figure 4) obtained from both the UV-VIS spectra and the Dentcolor program, were significantly different for all four types of nanocomposites stored in the coffee; while no significant difference was observed in those stored in artificial saliva.

Natural teeth emit strong blue fluorescence under the action of ultraviolet light [41] Ideal aesthetic restoration materials should have fluorescence similar to that of teeth [42]. If the fluorescence emission of a composite resin is increased, a higher brightness effect can be obtained [43]. The basic components of the composite resins do not favor fluorescence. Dental substrates emit fluorescence due to the presence of amino acids, such as tryptophan, present in the human dentin [44]. From the examination of emission spectra (Figure 5), it can be observed that the bleached composites have an intense emission band at 454 nm.

The Nanofil composite (Schulzer) has higher fluorescence intensity after bleaching with experimental Gel G28, followed by P1 after G28 gel bleaching. The initial P2 composite has the weakest fluorescence intensity. The fluorescence emmisions of the three experimental composites before and after bleaching, has different intensities compared to the commercial composite. In the literature data, when the human dentin was irradiated with 365 nm UV light, fluorescence was observed with a peak at 440 ± 10 nm [45].

Microstructural analysis was conducted, aiming to obtain information on the effects that the bleaching agents have on the surface of the dental restoration materials. The influence of the different bleaching agents on physical properties, surface and color morphology for different restorative materials has been investigated in several in vitro studies [46] that simulate a clinical situation as closely as possible. Currently, there are both positive and negative studies regarding the action of bleaching agents on restorative materials used in the oral cavity. Some recent studies have shown that they can affect the surface of the restoration materials, but all of these studies have been carried out on synthetic, and not on natural, bleaching agents [47]. The behavior of the bleaching effect on the composite surface and structure differs, depending on the method of sample processing and on the composition of the composite material.

With cytotoxicity assays, the cellular response can be described morphologically or quantitatively based on the viability, proliferation and cell function. To obtain a good contact between the cells and the investigated materials, the cells can be grown directly on their surface. In this case, the surface characteristics of materials are of particular importance, because if the material has low surface energy, the cells will not adhere to the surface of the material, and therefore will not grow well [48,49].

The viability of untreated cultures increased significantly with the exposure time, for only 24 h (*p* = 0.018), a marker of cell proliferation. As seen in Figure 7, the compounds studied had a low cytotoxic effect on cell cultures. The most important effect of decreasing the viability was recorded for commercial carbamide GC (Opalescence 15%) on fibroblasts. G28 experimental gel had the lowest cytotoxic effect.

In the case of the composites subjected to the whitening treatment, the analysis indicated that the studied compounds had a low cytotoxic effect on cell cultures. The bleached composite samples exposure was tolerated well by the cells. Only the P11 sample significantly diminished the cell viability after 24 h exposure, while at 72 h, this effect was not seen. P31 and NS seemed to have a stimulating effect on the cell viability, while P61 had no significant effect (Figure 8).

## 5. Conclusions

The organic acids content in studied bleaching gels ranges between 19.3294 mg/g for GEL 29 (based on quince natural juice) and 39.8629 mg/g for GEL 28 (based on natural pineapple and quince juices. By examining the emission spectra, it could be observed that the bleached composites had an intensive emission band at 454 nm. All of the samples of nanocomposites immersed in coffee showed the most significant variance in color compared to the initial values. The interaction between immersion time and the type of bleaching gel was significant. Both the ΔE values obtained from the UV-VIS spectra and the Dentcolor program were significantly different for all four types of nanocomposites stored in the coffee, while no noteworthy difference was observed in those stored in artificial saliva. The surface of the investigated samples differs depending on the type of gel, the type of composite and the degree of surface processing. The studied gels had a low cytotoxic effect on cell cultures. The lowest cytotoxic effect is held by the G28 experimental gel. Gels have acted differently on the different studied composites. Experimental natural gel bleaching treatments can lead to a reduction in coloring, without making any changes in surface morphology of dental composites.

## Figures and Tables

**Figure 1 materials-12-02106-f001:**
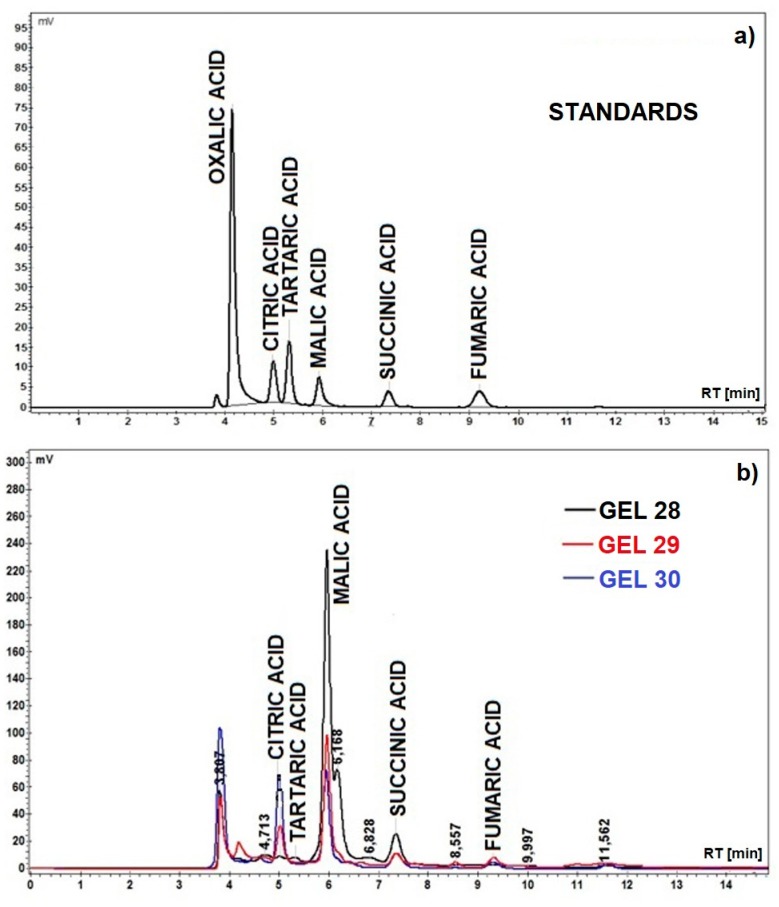
(**a**) Chromatogram of the standard mixture (1 mg/mL) of studied organic acids: oxalic acid (RT = 4.145 min), citric acid (RT = 4.987 min), tartaric acid (RT = 5.310 min), malic acid (RT = 5.925 min), succinic acid (RT = 7.350 min) and fumaric acid (RT = 9.207 min); (**b**) Chromatograms of the studied experimental bleaching gels.

**Figure 2 materials-12-02106-f002:**
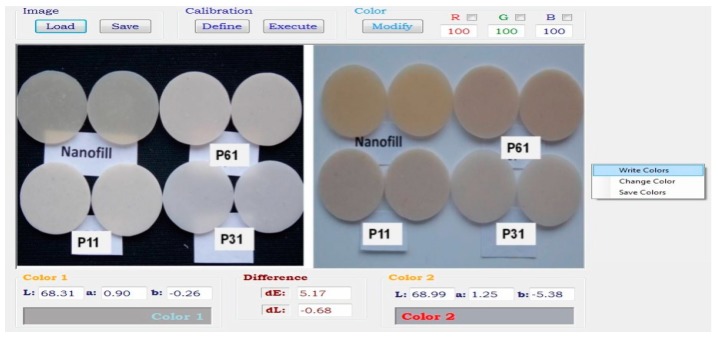
Digital images used for color difference (ΔE_ab_) by applying the original software (Dentcolor application), initial and after coloring.

**Figure 3 materials-12-02106-f003:**
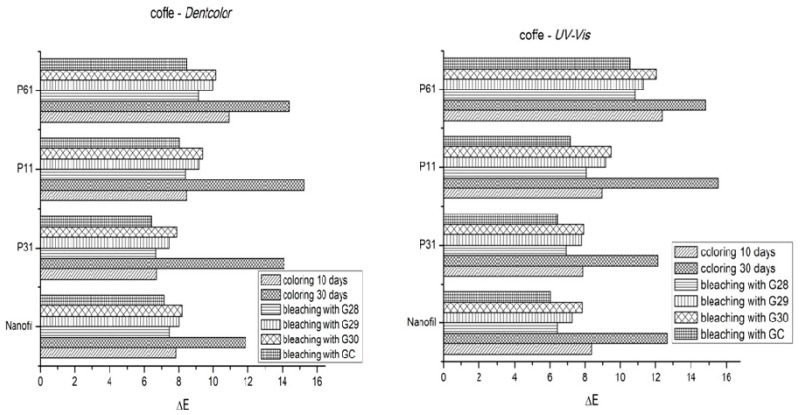
ΔE_ab_ color difference values for experimental composite samples (P31, P11, P61) and commercial composite “Nanofill”, colored in “coffee” for 10 and 30 days, respectively, and bleached with natural gels G28, G29, G30 and comercial gel GC (Opalescence 15%), recorded by: (**a**) DENTCOLOR program with digitized images; (**b**) UV-VIS spectrophotometer.

**Figure 4 materials-12-02106-f004:**
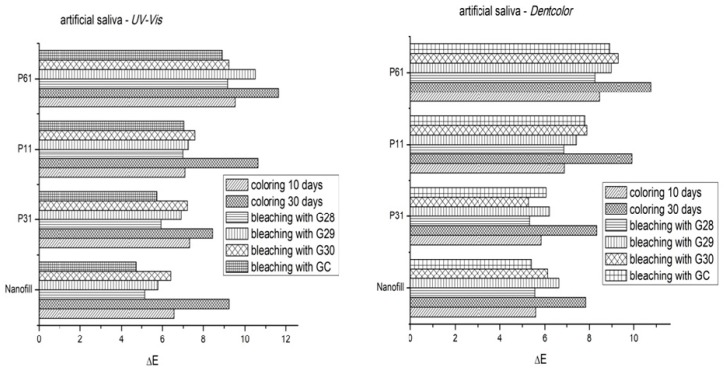
The color difference values ΔE for experimental composite samples and commercial composite, colored in ARTIFICIAL SALIVA than bleached with natural gels G28, G29, G30 and GC (Opalescence 15%) commercial gel, recorded by: (**a**) DENTCOLOR program with digitized images; (**b**) UV-VIS spectrophotometer.

**Figure 5 materials-12-02106-f005:**
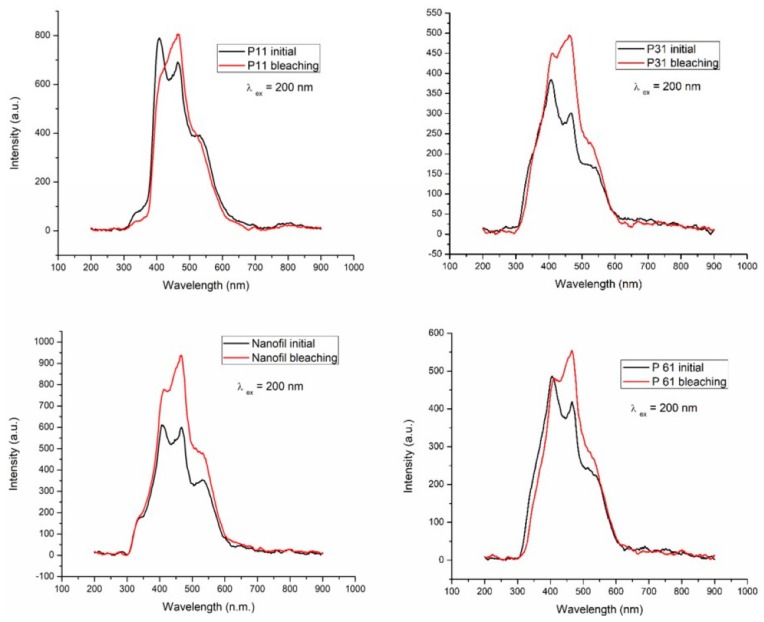
The emission spectrum of the four composites before and after bleaching with the G28 experimental gel.

**Figure 6 materials-12-02106-f006:**
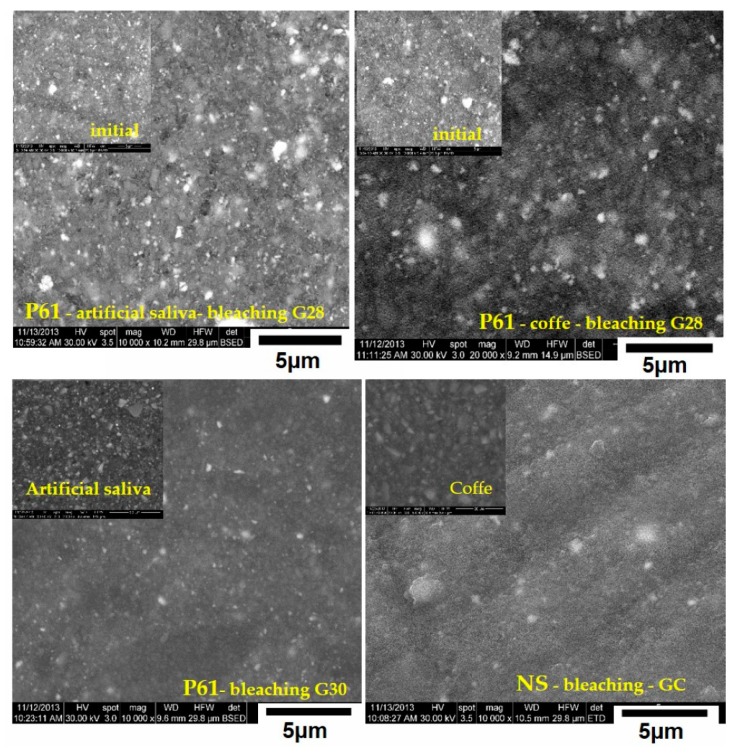
SEM images for experimental composite materials (**a**) P11—image left top of the original sample, large image is G28 bleached composite after storage in artificial saliva for 30 days; (**b**) P61—image left top of the original sample, large image is G29 bleached composite after storage in coffee; (**c**) P61—image the left top of the original sample, large image is G30 bleached composite after storage in artificial saliva; (**d**) Nanofill—image in the left top of the original sample, large image is GC commercial gel bleached composite after storage in the coffee.

**Figure 7 materials-12-02106-f007:**
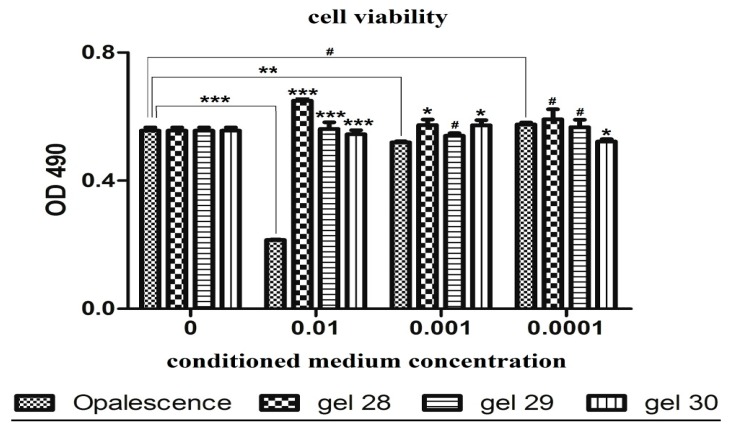
Cell viability for the four tested gels as a function of gel concentration in the conditioned medium comparative diagrams of cell viability of dental follicle stem cells following exposure to Opalescence, experimental gels 28, 29 and 30 (0.01–0.0001 dilutions of the conditioned medium) (upper panel); Each bar represents mean ± standard deviation (n = 3); # not significant, * *p* < 0.05; ** *p* < 0.01, *** *p* < 0.001, compared to Opalescence (upper panel).

**Figure 8 materials-12-02106-f008:**
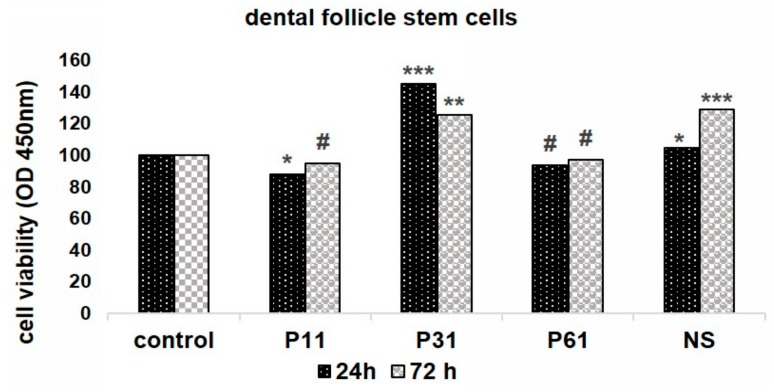
Cell viability of dental follicles on bleached cement samples exposure (lower panel) with G28 bleaching gel; each bar represents mean ± standard deviation (n = 3); # not significant, * *p* < 0.05; ** *p* < 0.01, *** *p* < 0.001, compared to untreated controls, (upper panel).

**Table 1 materials-12-02106-t001:** Composition of investigated composite materials.

Composite	Organic Matrix	Fillers	Ratio L/F [wt]
P11 *	Bis-GMA*, TEGDMA, UDMA	Glass with BaF_2_ *, colloidal silica, HA-Zn *	23/77
P31 *	Bis-GMA*, TEGDMA, UDMA	Glass with BaO *, Mixture of oxides *, Colloidal silica	23/77
P61 *	Bis-GMA*, TEGDMA, UDMA	Glass with Sr si Zr *, SiO_2_-Zr *, HA-F *, HA-Zr *	21.8/78.19
Nanofill	UDMA-resin	0.1–2 μm glass and 100 nm nanoparticles	18/82

Notes: UBB-ICCRR: Babes-Bolyai University, Institute of Chemistry Raluca Ripan, Cluj-Napoca Romania; *Bis-GMA: 2,2-Bis[p-(2-hydroxy-3-methacryloyloxypropoxy)-phenyl]-propane (UBB-ICCRR); TEGDMA: triethylene glycol dimethacrylate (Aldrich); UDMA: urethane dimetacrylate (Aldrich); Colloidal silica (Degussa, Germany); * fillers synthetized in UBB-ICCRR laboratory; L/F: liquid organic matrix/fillers.

**Table 2 materials-12-02106-t002:** Content of organic acids (mg/g gel) in bleaching gels from HPLC analysis.

GEL Code	Oxalic Acid	Citric Acid	Tartaric Acid	Malic Acid	Succinic Acid	Fumaric Acid	Total Acidity
GEL 28	-	0.132	0.124	33.39	6.216	0.0009	39.8629
GEL 29	0.0165	2.608	-	14.146	2.55	0.0089	19.3294
GEL 30	-	6.77	-	9.859	3.73	0.0066	20.3656

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
