# Peer review of "Preparation and Characterization of Natural Bleaching Gels Used in Cosmetic Dentistry"

_materials, 2019, doi:10.3390/ma12132106_

Round 1
Reviewer 1 Report
The manuscript entitled "Preparation and characterization of natural bleaching 2 gels used in cosmetic dentistry" by Mazilu et al. is interesting. Major revision is suggested.
In the text: mg/mL should be without space before and after slash, /.
There is a scheme before materials and methods. What is this? it doesn't have caption. in addition, it should be enlarged.
For all figures, please do not use "time new Roman" font. Arial is more readable.
Fig. 2 does not have good quality.
Fig. 3 and 4 may be re-plotted with a better software than Excel such as origin pro. Also fig 7 is not good.
Some fig captions such as 7 and 8 should contain information about p value that used for the graph.
Author Response
Point 1. In the text: mg/mL should be without space before and after slash, /.
Response 1. The spaces after and before slash were removed.
Point 2. There is a scheme before materials and methods. What is this? it doesn't have caption. in addition, it should be enlarged
Response 2. The scheme before materials and methods with no number is a graphical abstract.
Point 3. For all figures, please do not use "time new Roman" font. Arial is more readable.
Fig. 2 does not have good quality.
Fig. 3 and 4 may be re-plotted with a better software than Excel such as origin pro. Also fig 7 is not good.
Response 3. We improved the figures in order to increase the clarity and to be more readable.
Point 4. Some fig captions such as 7 and 8 should contain information about p value that used for the graph.
Response 4. For figures 7 and 8 we introdued information about p value that used for the graph into the legend of the figure.
Reviewer 2 Report
The authors of "Preparation and characterization of natural bleaching gels used in cosmetic dentistry" study the effect of various bleaching agents on the surface and color morphology of different composite materials and experimental nanocomposites and the cytotoxicity assay (on gels and nanocomposites after bleaching. The topic is of interest, due to the potential commercial impact in the dental products for whitening the teeth as well as for other applications.
The manuscript is relatively well written, though there is considerable room for improvement both in the writing and in the presentation of the results. The methods and approaches seems sound to me, though I am not an expert on these techniques. Overall, I will recommend the manuscript provided the authors make the following major improvements:
1) The figures need significant improvement, in terms in resolution, clarity, and organization. Figures 1 & 2 require immediate improvement.
2) Figure 5 is very crowed. If printed in black and white, then it will become very hard to the reader to distinguish the various line. I strongly suggest to the authors to break the figure into subfigures so that the reader more easily appreciate the results.
3) The authors multiple time make various statements without providing references, I include a couple examples that caught my eye:
The authors at page 1: "Throughout the world, many studies are conducted on the analysis...", can the authors provide notable example for these studies?
page 2: "The use of the natural products in dentistry was indicated in various situations," can the authors provide a specific examples with references.
4) The discussion section is not well written, the authors need to re-write the particular section by avoiding paragraph fragmentation. Avoid details and present the gist of their findings.
5) It would be interesting if the authors can comments about the potential impact of the substances that they examine on potential impact on gum and penetration to the blood stream via the blood gum vessels.
Author Response
Point 1. The figures need significant improvement, in terms in resolution, clarity, and organization. Figures 1 & 2 require immediate improvement.
Point 2. Figure 5 is very crowed. If printed in black and white, then it will become very hard to the reader to distinguish the various line. I strongly suggest to the authors to break the figure into subfigures so that the reader more easily appreciate the results.
Response 1, 2. We improved the figures in terms in resolution, clarity, and organization and modified figures 1, 2, 3, 4, 5,6,7,8.
Point 3. The authors multiple time make various statements without providing references, I include a couple examples that caught my eye:
The authors at page 1: "Throughout the world, many studies are conducted on the analysis...", can the authors provide notable example for these studies?
page 2: "The use of the natural products in dentistry was indicated in various situations," can the authors provide a specific examples with references.
Response 3. We introduced new references for a series of statements from the Introduction.
Point 4. The discussion section is not well written, the authors need to re-write the particular section by avoiding paragraph fragmentation. Avoid details and present the gist of their findings
.
Response 4. We tried to improve the discussion section.
Point 5. It would be interesting if the authors can comments about the potential impact of the substances that they examine on potential impact on gum and penetration to the blood stream via the blood gum vessels.
Response 5. We made no comments on the potential impact of the substances that they examine on potential impact on gum and penetration to the blood stream via the blood gum vessel because the short time available for revision, but we showed that our gels presents no toxicity and no toxic substances were used for gel preparation.
Reviewer 3 Report
The manuscript “Preparation and characterization of natural bleaching gels used in cosmetic dentistry” was not well prepared.
First of all, the authors needs extensive editing to improve grammar and correct spell as well as the writing to make their points clear.
Also, quality of all figures need to be improved, e.g., include error bars, clearly show x/y-axis legends, organize and display before and after SEM images for easier comparison.
HPLC experimental details is missing: mobile phase, running condition, column type etc?
Author Response
Point 1. First of all, the authors needs extensive editing to improve grammar and correct spell as well as the writing to make their points clear.
Response 1. We tried to improve grammar and correct the spell of the manuscript.
Point 2. Also, quality of all figures need to be improved, e.g., include error bars, clearly show x/y-axis legends, organize and display before and after SEM images for easier comparison.
Response 2. We improved the quality of the figures and completed the legends of the figures.
Point 3. HPLC experimental details is missing: mobile phase, running condition, column type etc?
Response 3. We introduced experimental details related to HPLC experiment.
Reviewer 4 Report
The paper is interesting and the manuscript is well performed. However some minor concerns should be addressed in order to have a more strong paper. Introduction section is weak and some data about the influence of no natural product over the teeth and saliva should be added in order to have a research directed not to just scientist but also to the clinicians
Paper like the following
Villa er al. J Am Dent Assoc. 2011 Jul;142(7):811-6. Dental patients self reports of xerostomia and associated risk factors
Could be added in order to increase that section
Overall an interesting paper
Author Response
Point 1. The paper is interesting and the manuscript is well performed. However some minor concerns should be addressed in order to have a more strong paper. Introduction section is weak and some data about the influence of no natural product over the teeth and saliva should be added in order to have a research directed not to just scientist but also to the clinicians
Paper like the following
Villa er al. J Am Dent Assoc. 2011 Jul;142(7):811-6. Dental patients self reports of xerostomia and associated risk factors
Could be added in order to increase that section
Response 1. We improved the introduction based on new references, including some paper that approaches the impact of non-natural products on bovine teets and xerostomia. Unfortunately we have no acces to the full paper related to xerostomia, and consulted only the abstract freely available in the Internet.
Round 2
Reviewer 2 Report
The authors addressed my comments. I approve the manuscript for publication.
Author Response
Dear Editor,
Thank you for approve the manuscript for publication.
As for our update on the revisions, we kindly inform you that I have attached the manuscript with minor revisions of English.
All revisions have been corrected and clearly highlighted, using the "Track Changes" function in Microsoft Word, so that they are easily visible.
Best regards,
Dr. Doina Prodan
